# OPEN RL BENCHMARK: Comprehensive Tracked Experiments for Reinforcement Learning

**Shengyi Huang**[1,2*]   **Quentin Gallouédec**[1,3*]   **Florian Felten**[4]   **Antonin Raffin**[5]
**Rousslan Fernand Julien Dossa**[6]   **Yanxiao Zhao**[7,8]   **Ryan Sullivan**[9]   **Viktor Makoviychuk**[10]
**Denys Makoviichuk**[11]   **Mohamad H. Danesh**[12]   **Cyril Roumégous**[13]   **Jiayi Weng**
**Chufan Chen**[14]   **Md Masudur Rahman**[15]   **João G. M. Araújo**[16]   **Guorui Quan**[17]
**Daniel C.H. Tan**[18,19]   **Timo Klein**[20,21]   **Rujikorn Charakorn**[22]   **Mark Towers**[23]
**Yann Berthelot**[24,25]   **Kinal Mehta**[26]   **Dipam Chakraborty**[27]   **Arjun KG**
**Valentin Charraut**[28]   **Chang Ye**[29]   **Zichen Liu**[30]   **Lucas N. Alegre**[31]   **Alexander Nikulin**[32]
**Xiao Hu**[33]   **Tianlin Liu**[34]   **Jongwook Choi**[35]   **Brent Yi**[36]

## Abstract

In many Reinforcement Learning (RL) papers, learning curves are useful indicators to measure the effectiveness of RL algorithms. However, the complete raw data of the learning curves are rarely available. As a result, it is usually necessary to reproduce the experiments from scratch, which can be time-consuming and error-prone. We present OPEN RL BENCHMARK (ORLB), a set of fully tracked RL experiments, including not only the usual data such as episodic return, but also all algorithm-specific and system metrics. ORLB is community-driven: anyone can download, use, and contribute to the data. At the time of writing, more than 25,000 runs have been tracked, for a cumulative duration of more than 8 years. It covers a wide range of RL libraries and reference implementations. Special care is taken to ensure that each experiment is precisely reproducible by providing not only the full parameters, but also the versions of the dependencies used to generate it. In addition, ORLB comes with a command-line interface (CLI) for easy fetching and generating figures to present the results. In this document, we include two case studies to demonstrate the usefulness of ORLB in practice. To the best of our knowledge, ORLB is the first RL benchmark of its kind, and the authors hope that it will improve and facilitate the work of researchers in the field.

## 1   Introduction

Reinforcement Learning (RL) research is based on comparing new methods to baselines to assess progress (Patterson et al., 2023). This process requires the availability of the data associated with these baselines (Raffin et al., 2021) or, alternatively, the ability to replicate them and generate the data oneself (Raffin, 2020). In addition, reproducible results allow the methods to be compared with new benchmarks and to identify the areas in which the methods excel and those in which they are likely to fail, thus providing avenues for future research.

In practice, the RL research community faces complex challenges in comparing new methods with reference data. The unavailability of reference data requires researchers to reproduce experiments, which is difficult due to insufficient source code documentation and evolving software dependencies.

---

[*]Equal contributions

Submitted to the 38th Conference on Neural Information Processing Systems (NeurIPS 2024) Track on Datasets and Benchmarks. Do not distribute.

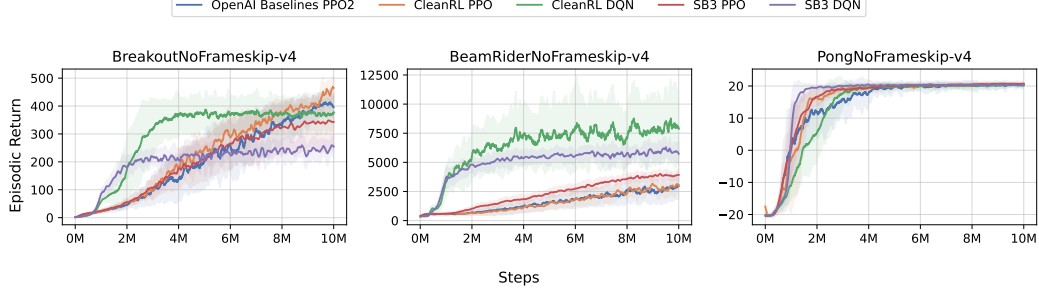

Figure 1: Example of learning curves obtained with OPEN RL BENCHMARK. These compare the episodic returns obtained by different implementations of PPO and DQN on three Atari games.

Implementation details, as highlighted in past research, can significantly impact results (Henderson et al., 2018; Huang et al., 2022a). Moreover, limited computing resources play a crucial role, hindering the reproduction process and affecting researchers without substantial access.

The lack of standardized metrics and benchmarks across studies not only impedes comparison but also results in a substantial waste of time and resources. To address these issues, the RL community must establish rigorous reproducibility standards, ensuring replicability and comparability across studies. Transparent sharing of data, code, and experimental details, along with the adoption of consistent metrics and benchmarks, would collectively enhance the evaluation and progression of RL research, ultimately accelerating advancements in the field.

ORLB presents a rich collection of tracked RL experiments and aims to set a new standard by providing a diverse training dataset. This initiative prioritizes the use of existing data over re-running baselines, emphasizing reproducibility and transparency. Our contributions are:

- **Extensive dataset:** Offers a large, diverse collection of tracked RL experiments.
- **Standardization:** Establishes a new norm by encouraging reliance on existing data, reducing the need for re-running baselines.
- **Comprehensive metrics:** Includes diverse tracked metrics for method-specific and system evaluation, in addition to episodic return.
- **Reproducibility:** Emphasizes clear instructions and fixed dependencies, ensuring easy experiment replication.
- **Resource for research:** Serves as a valuable and collaborative resource for RL research.
- **Facilitating exploration:** Enables reliable exploration and assessment of new and exisiting RL methods.

## 2 Comprehensive overview of ORLB: content, methodology, tools, and applications

This section provides a detailed exploration of the contents of ORLB, including its diverse set of libraries and environments, and the metrics it contains. We also look at the practical aspects of using ORLB, highlighting its ability to ensure accurate reproducibility and facilitate the creation of data visualizations thanks to its CLI.

### 2.1 Content

ORLB data is stored and shared with Weights and Biases (Biewald, 2020). The data is contained in a common entity named `openrlbenchmark`. Runs are divided into several *projects*. A project can correspond to a library, but it can also correspond to a set of more specific runs, such as `envpool-cleanrl` in which we find CleanRL runs (Huang et al., 2022b) launched with the EnvPool

implementation of environments (Weng et al., 2022b). A project can also correspond to a reference implementation, such as TD3 (project `sfujim-TD3`) or Phasic Policy Gradient (Cobbe et al., 2021) (project `phasic-policy-gradient`). ORLB also includes reports, which are interactive documents designed to enhance the visualization of selected representations. These reports provide a more user-friendly format for practitioners to share, discuss, and analyze experimental results, even across different projects. Figure 2 shows a preview of one such report.

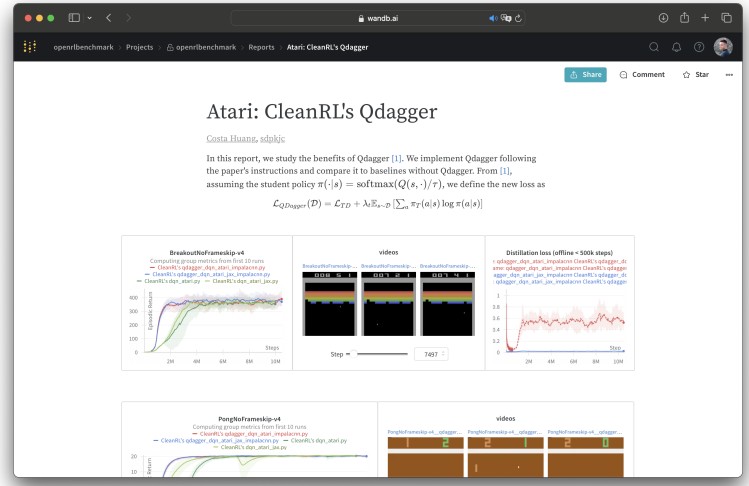

Figure 2: An example of a report on the Weights and Biases platform, dealing with the contribution of QDagger (Agarwal et al., 2022), and using data from ORLB. The URL to access the report is `https://wandb.ai/openrlbenchmark/openrlbenchmark/reports/Atari-CleanRL-s-Qdagger--VmlldzoONTg1ODY5`.

At the time of writing, ORLB contains nearly 25,000 runs, for a total of 72,000 hours (more than 8 years) of tracking. In the following paragraphs, we present the libraries and environments for which runs are available in ORLB, as well as the metrics tracked.

**Libraries**    ORLB contains runs for several reference RL libraries. These libraries are: abcdRL (Zhao, 2022), Acme (Hoffman et al., 2020), Cleanba (Huang et al., 2023), CleanRL (Huang et al., 2022b), jaxrl (Kostrikov, 2021), moolib (Mella et al., 2022), MORL-Baselines (Felten et al., 2023), OpenAI Baselines (Dhariwal et al., 2017), rlgames (Makoviichuk & Makoviychuk, 2021) Stable Baselines3 (Raffin et al., 2021; Raffin, 2020) Stable Baselines Jax (Raffin et al., 2021) and TorchBeast (Küttler et al., 2019).

**Environments**    The runs contained in ORLB cover a wide range of classic environments. They include Atari (Bellemare et al., 2013; Machado et al., 2018), Classic control (Brockman et al., 2016), Box2d (Brockman et al., 2016) and MuJoCo (Todorov et al., 2012) as part of either Gym (Brockman et al., 2016) or Gymnasium (Towers et al., 2023) or EnvPool (Weng et al., 2022b). They also include Bullet (Coumans & Bai, 2016), Procgen Benchmark (Cobbe et al., 2020), Fetch environments (Plappert et al., 2018), PandaGym (Gallouédec et al., 2021), highway-env (Leurent, 2018), Minigrid (Chevalier-Boisvert et al., 2023) and MO-Gymnasium (Alegre et al., 2022).

**Tracked metrics**    Metrics are recorded throughout the learning process, consistently linked with a global step indicating the number of interactions with the environment, and an absolute time, which allows to compute the duration of a run. We categorize these metrics into four distinct groups:

- **Training-related metrics:** These are general metrics related to RL learning. This category contains, for example, the average returns obtained, the episode length or the number of collected samples per second.

- **Method-specific metrics:** These are losses and measures of key internal values of the methods. For PPO, for example, this category includes the value loss, the policy loss, the entropy or the approximate KL divergence.

- **Evolving configuration parameters:** These are configuration values that change during the learning process. This category includes, for example, the learning rate when there is decay, or the exploration rate ($\epsilon$) in the Deep Q-Network (DQN) (Mnih et al., 2013).

- **System metrics:** These are metrics related to system components. These could be GPU memory usage, its power consumption, its temperature, system and process memory usage, CPU usage or even network traffic.

The specific metrics available may vary from one library to another. In addition, even where the metrics are technically similar, the terminology or key used to record them may vary from one library to another. Users are advised to consult the documentation specific to each library for precise information on these measures.

## 2.2 Everything you need for perfect repeatability

Reproducing experimental results in computational research, as discussed in Section 4.3, is often challenging due to evolving codebases, incomplete hyperparameter listings, version discrepancies, and compatibility issues. Our approach aims to enhance reproducibility by ensuring users can exactly replicate benchmark results. Each experiment includes a complete configuration with all hyperparameters, frozen versions of dependencies, and the exact command, including the necessary random seed, for systematic reproducibility. As a example, CleanRL (Huang et al., 2022b) introduces a unique utility that streamlines the process of experiment replication (see Figure 3). This tool produces the command lines to set up a Python environment with the necessary dependencies, download the run file, and the precise command required for the experiment reproduction. Such an approach to reproduction facilitates research and makes it possible to study in depth unusual phenomena, or cases of rupture[2], in learning processes, which are generally ignored in the results presented, either because they are deliberately left out or because they are erased by the averaging process.

```
$ python -m cleanrl_utils.reproduce --run openrlbenchmark/cleanrl/runs/c1y1qnz4

# run the following
python3 -m venv venv
source venv/bin/activate
pip install -r https://api.wandb.ai/files/openrlbenchmark/cleanrl/c1y1qnz4/requirements.txt
curl -OL https://api.wandb.ai/files/openrlbenchmark/cleanrl/c1y1qnz4/code/cleanrl/ppo_atari.py
python ppo_atari.py --track --env-id BreakoutNoFrameskip-v4 --seed 3
```

Figure 3: CleanRL's module `reproduce` allows the user to generate, from an ORLB run reference, the exact command suite for an identical reproduction of the run.

## 2.3 The CLI for generating figures in one command line

ORLB offers convenient access to raw data from RL libraries on standard environments. It includes a feature for easily extracting and visualizing data in a paper-friendly format, streamlining the process of filtering and extracting relevant runs and metrics for research papers through a single command. The CLI is a powerful tool for generating most metrics-related figures for RL research and notably, all figures in this document were generated using the CLI. The data in ORLB can also be accessed by custom scripts, as detailed in Appendix A.2. Specifically, the CLI integrated into ORLB provides users with the flexibility to:

---

[2]Exemplified in `https://github.com/DLR-RM/rl-baselines3-zoo/issues/427`

- Specify algorithms' implementations (from which library) along with their corresponding git commit or tag;

- Choose target environments for analysis;

- Define the metrics of interest;

- Opt for the additional generation of metrics and plots using RLiable (Agarwal et al., 2021).

Concrete example usage of the CLI and resulting plots are available in Appendix A.1.

## 3 ORLB in action: an insight into case studies

ORLB offers a powerful tool for researchers to evaluate and compare different RL algorithms. In this section, we will explore two case studies that showcase its benefits. First, we propose to investigate the effect of using TD($\lambda$) for value estimation in PPO (Schulman et al., 2017) versus using Monte Carlo (MC). This simple study illustrates the use of ORLB through a classic research question. Moreover, to the best of our knowledge, this question has never been studied in the literature. We then show how ORLB is used to demonstrate the speedup and variance reduction of a new IMPALA implementation proposed by Huang et al. (2023). By using ORLB, we can save time and resources while ensuring consistent and reproducible comparisons. These case studies highlight the role of the benchmark in providing insights that can advance the field of RL research.

### 3.1 Easily assess the contribution of TD($\lambda$) for value estimation in PPO

In the first case study, we show how ORLB can be used to easily compare the performance of different methods for estimating the value function in PPO (Schulman et al., 2017), one of the many implementation details of this algorithm (Huang et al., 2022a). Specifically, we compare the commonly used Temporal Difference (TD)($\lambda$) estimate to the Monte-Carlo (MC) estimate.

PPO typically employs Generalized Advantage Estimation (GAE) (Schulman et al., 2016) to update the actor. The advantage estimate is expressed as follows:

$$A_t^{\text{GAE}(\gamma,\lambda)} = \sum_{l=0}^{N-1} (\gamma\lambda)^l \delta_{t+l}^V \tag{1}$$

where $\lambda \in [0, 1]$ adjusts the bias-variance tradeoff and $\delta_{t+l}^V = R_{t+l} + \gamma\hat{V}(S_{t+l+1}) - \hat{V}(S_{t+l})$. The target return for critic optimization is estimated with TD($\lambda$) as follows:

$$G_t^\lambda = (1 - \lambda) \sum_{n=1}^{\infty} \lambda^{n-1} G_{t:t+n} \tag{2}$$

where $G_{t:t+n} = \sum_{k=0}^{n-1} \gamma^k R_{t+k+1} + \gamma^n V(S_{t+n})$ is the $n$-steps return. In practice, the target return for updating the critic is computed from the GAE value, by adding the minibatch return, a detail usually overlooked by practitioners (Huang et al., 2022a, point 5). While previous studies (Patterson et al., 2023) have shown the joint benefit of GAE and over MC estimates for actor and critic, we focus on the value function alone. To isolate the influence of the value function estimation, we vary the method used for the value function and keep GAE for advantage estimation.

The first step is to identify the reference runs in ORLB. Since PPO is a well-known baseline, there are many runs available; we decided to use those from Stable Baselines3 for this example. We then retrieve the exact source code and command used to generate the runs – thanks to the pinned dependencies that come with them – and make the necessary changes to the source code. For each selected environment, we start three learning runs using the same command as the one we retrieved. The runs are saved in a dedicated project[3]. For fast and user-friendly rendering of the results, we

---

[3] https://wandb.ai/modanesh/openrlbenchmark

create a Weights and Biases report[4]. Using ORLB CLI, we generate Figure 4 and 5. The command used to generate the figures is given in Appendix B.

Figures 4 and 5 give an overview of the results, while detailed plots in the Appendix B provide a closer look at each environment. The proposed modification to the PPO value function estimation has an impact on the performance for Atari games (Figure 4a): not using TD($\lambda$) results in lower scores. However, PPO with MC estimates has similar performance to the original PPO in Box2D and MuJoCo environments. This example shows how ORLB can be used to quickly investigate the influence of design choices in RL. It provides baseline results and tools to compare and reproduce results.

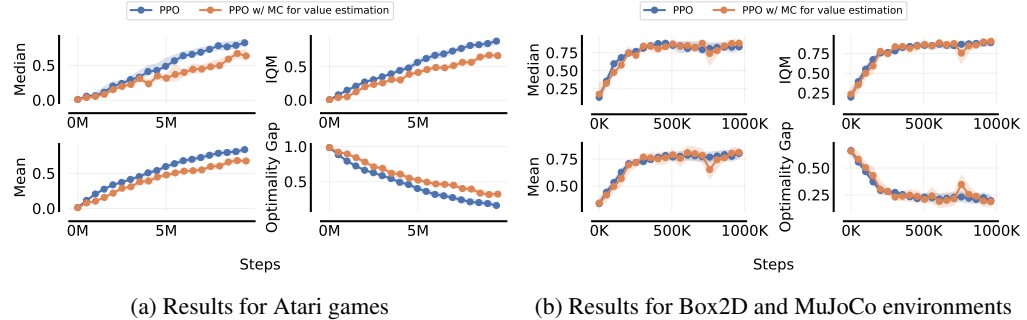

| (a) Results for Atari games | (b) Results for Box2D and MuJoCo environments |

Figure 4: Comparing the original PPO and the PPO with Monte-Carlo (MC) for value estimation. These experiments were conducted over 15 environments, including Atari games, Box2D, and MuJoCo. The plot shows min-max normalized scores with 95% stratified bootstrap CIs.

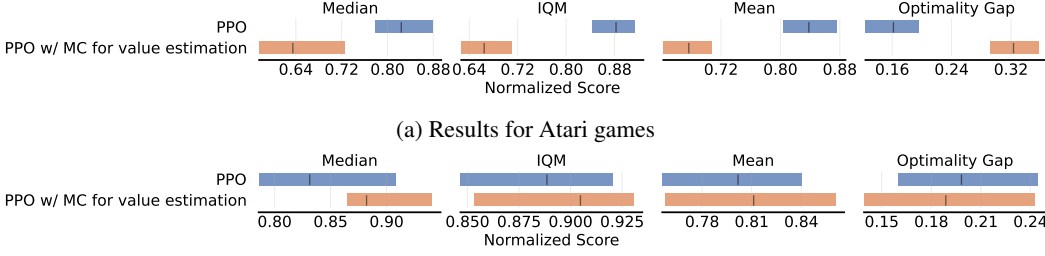

(a) Results for Atari games

(b) Results for Box2D and MuJoCo environments

Figure 5: Study of the contribution of GAE for estimating the value used to update the critic in PPO, compared against its variant which uses the MC estimator instead. Figures show the aggregated min-max normalized scores with stratified 95% stratified bootstrap CIs.

## 3.2  Demonstrating the utility of ORLB through the Cleanba case study

This section describes how ORLB was instrumental in the evaluation and presentation of Cleanba (Huang et al., 2023), a new open-source platform for distributed RL implementing highly optimized distributed variants of PPO (Schulman et al., 2017) and IMPALA (Espeholt et al., 2018). Cleanba's authors asserted three points: (1) Cleanba implementations compare favorably with baselines in terms of sample efficiency, (2) for the same system, the Cleanba implementation is more optimized and therefore faster, and (3) the design choices allow a reduction in the variability of results.

To prove these assertions, the evaluation of Cleanba encountered a common problem in RL research: the works that initially proposed these baselines did not provide the raw results of their experiments. Although a reference implementation is available[5], it is no longer maintained. Subsequent works such as Moolib (Mella et al., 2022) and TorchBeast (Küttler et al., 2019) have successfully replicated

---

[4]https://api.wandb.ai/links/modanesh/izf4yje4
[5]https://github.com/google-deepmind/scalable_agent

the IMPALA results. However, these shared results are limited to the paper's presented curves, which provide a smoothed measure of episodic return as a function of interaction steps on a specific set of Atari tasks. It is worth noting that these tasks are not an exact match for the widely recognized Atari 57, and the raw data used to generate these curves is unavailable.

Recognizing the lack of raw data for existing IMPALA implementations, the authors reproduced the experiments, tracked the runs and integrated them into ORLB. As a reminder, these logged data include not only the return curves, but also the system configurations and temporal data, which are crucial to support the Cleanba authors' optimization claim. Comparable experiments have been run, tracked and shared on ORLB with the proposed Cleanba implementation.

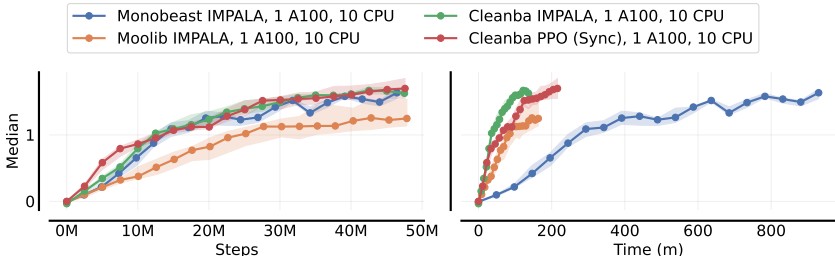

Figure 6: Median human-normalized scores with 95% stratified bootstrap CIs of Cleanba (Huang et al., 2023) variants compared with moolib (Mella et al., 2022) and monobeast (Küttler et al., 2019). The experiments were conducted on 57 Atari games (Bellemare et al., 2013). The data used to generate the figure comes from ORLB, and the figure was generated with a single command from ORLB's CLI. Figure from (Huang et al., 2023).

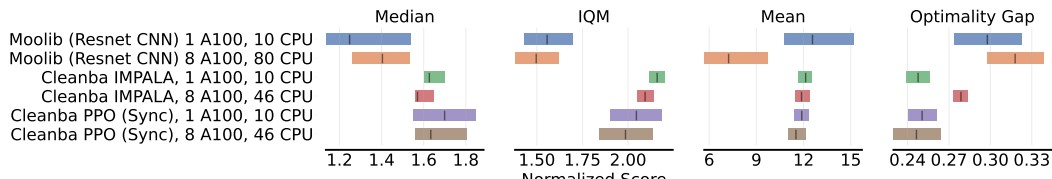

Figure 7: Aggregated normalized human scores with stratified 95% bootstrap CIs, showing that unlike moolib (Mella et al., 2022), Cleanba (Huang et al., 2023) variants have more predictable learning curves (using the same hyperparameters) across different hardware configurations. Figure from (Huang et al., 2023).

Using ORLB CLI, the authors generated several figures. In Figure 6, taken from (Huang et al., 2023), the authors show that the results in terms of sample efficiency compare favorably with the baselines, and that for the same system configuration, convergence was temporally faster with the proposed implementation, thus proving claims (1) and (2). Figure 7 demonstrates that Cleanba variants maintain consistent learning curves across different hardware configurations. Conversely, moolib's IMPALA shows marked variability in similar settings, despite identical hyperparameters, confirming the authors' third claim.

# 4   Current practices in RL: data reporting, sharing and reproducibility

Many new methods have emerged in recent years, with some becoming standard baselines, but current practices in the field make it challenging to interpret, compare, and replicate study results. In this section, we highlight the inconsistent presentation of results, focusing on learning curves as an example. This inconsistency can hinder interpretation and lead to incorrect conclusions. We also note the insufficient availability of learning data, despite some positive efforts, and examine challenges related to method reproducibility.

## 4.1 Analyzing learning curve practices

Plotting learning curves is a common way to show an agent's performance over learning. We closely examine the components of learning curves and the choices made by key publications. We find a lack of uniformity, with presentation choices rarely explained and sometimes not explicitly stated.

**Axis**   Typically, the $y$ axis measures either the return acquired during data collection or evaluation. Some older papers, like (Schulman et al., 2015; Mnih et al., 2016; Schulman et al., 2017), fail to specify the metric, using the vague term *learning curve*. The first approach sums the rewards collected during agent rollout (Dabney et al., 2018; Burda et al., 2019). The second approach suspends training, averaging the agent's return over episodes, deactivating exploration elements (Fujimoto et al., 2018; Haarnoja et al., 2018; Hessel et al., 2018; Janner et al., 2019; Badia et al., 2020b; Ecoffet et al., 2021; Chen et al., 2021). This method is prevalent and provides a more precise evaluation. Regarding the $x$ axis, while older baselines (Schulman et al., 2015; Mnih et al., 2016) use policy updates and learning epochs, the norm is to use interaction counts with the environment. In Atari environments, it is often the number of frames, adjusting for frame skipping to match human interaction frequency.

**Shaded area**   Data variability is typically shown with a shaded area, but its definition varies across studies. Commonly, it represents the standard deviation (Chen et al., 2021; Janner et al., 2019) and less commonly half the standard deviation (Fujimoto et al., 2018). Haarnoja et al. (2018) uses a min-max representation to include outliers, covering the entire observed range. This method offers a comprehensive view but amplifies outliers' impact with more runs. Ecoffet et al. (2021) adopts a probabilistic approach, showing a 95% bootstrap confidence interval around the mean, ensuring statistical confidence. Unfortunately, Schulman et al. (2015, 2017); Mnih et al. (2016); Dabney et al. (2018); Badia et al. (2020b) omit statistical details or even the shaded area, introducing uncertainty in data variability interpretation, as seen in (Hessel et al., 2018).

**Normalization and aggregation**   Performance aggregation assesses method results across various tasks and domains, indicating their generality and robustness. Outside the Atari context, aggregation practices are uncommon due to the lack of a universal normalization standard. Without a widely accepted normalization strategy, scores are typically not aggregated, or if they are, it relies on a min-max approach lacking absolute significance and unsuitable for comparisons. Early Atari research did not use normalization or aggregate results (Mnih et al., 2013). There has been a shift towards normalizing against human performance, though this has weaknesses and may not reflect true agent mastery (Toromanoff et al., 2019). Aggregation methods vary: the mean is common but influenced by outliers, leading some studies to prefer the more robust median, as in (Hessel et al., 2018). Many papers now report both mean and median results (Dabney et al., 2018; Hafner et al., 2023; Badia et al., 2020a). Recent approaches, like using the Interquartile Mean (IQM), provide a more accurate performance representation across diverse games (Lee et al., 2022), as suggested by Agarwal et al. (2021).

## 4.2 Spectrum of data sharing practices

While the mentioned studies often have reference implementations (see Section 4.3), the sharing of training data typically extends only to the curves presented in their articles. This necessitates reliance on libraries that replicate these methods, offering benchmarks with varying levels of completeness. Several widely-used libraries in the field provide high-level summaries or graphical representations without including raw data (e.g., Tensorforce (Kuhnle et al., 2017), Garage (garage contributors, 2019), ACME (Hoffman et al., 2020), MushroomRL (D'Eramo et al., 2021), ChainerRL (Fujita et al., 2021), and TorchRL (Bou et al., 2023)). Spinning Up (Achiam, 2018) offers partial data accessibility, providing benchmark curves but withholding raw data. TF-Agent (Guadarrama et al., 2018) is slightly better, offering experiment tracking with links to TensorBoard.dev, though its future is uncertain due to service closure. Tianshou (Weng et al., 2022a) provides individual run reward data for Atari and average rewards for MuJoCo, with more detailed MuJoCo data available via a Google

Drive link, but it is not widely promoted. RLLib (Liang et al., 2018) maintains an intermediate stance in data sharing, hosting run data in a dedicated repository. However, this data is specific to select experiments and often presented in non-standard, undocumented formats, complicating its use. Leading effective data-sharing platforms include Dopamine (Castro et al., 2018) and Sample Factory (Petrenko et al., 2020). Dopamine consistently provides accessible raw evaluation data for various seeds and visualizations, along with trained agents on Google Cloud. Sample Factory offers comprehensive data via Weights and Biases (Biewald, 2020) and a selection of pre-trained agents on the Hugging Face Hub, enhancing reproducibility and collaborative research efforts.

### 4.3 Review on reproducibility

The literature shows variations in these practices. Some older publications like (Schulman et al., 2015, 2017; Bellemare et al., 2013; Mnih et al., 2016; Hessel et al., 2018) and even recent ones like (Reed et al., 2022) lack a codebase but provide detailed descriptions for replication[6]. However, challenges arise because certain hyperparameters, important but often unreported, can significantly affect performance (Andrychowicz et al., 2020). In addition, implementation choices have proven to be critical (Henderson et al., 2018; Huang et al., 2023, 2022a; Engstrom et al., 2020), complicating the distinction between implementation-based improvements and methodological advances.

Recognizing these challenges, the RL community is advocating for higher standards. NeurIPS, for instance, has been requesting a reproduction checklist since 2019 (Pineau et al., 2021). Recent efforts focus on systematic sharing of source code to promote reproducibility. However, codebases are often left unmaintained post-publication (with rare exceptions (Fujimoto et al., 2018)), creating complexity for users dealing with various dependencies and unsolved issues. To address these challenges, libraries have aggregated multiple baseline implementations (see Section 2.1), aiming to match reported paper performance. However, long-term sustainability remains a concern. While these libraries enhance reproducibility, in-depth repeatability is still rare.

## 5 Discussion and conclusion

Reproducing results in RL research is often difficult due to limited access to data and code, as well as the impact of minor implementation variations on performance. Researchers typically rely on imprecise comparisons with paper figures, making the reproduction process time-consuming and challenging. To address these issues, we introduce ORLB, a large collection of tracked experiments spanning various algorithms, libraries and benchmarks. ORLB records all relevant metrics and data points, offering detailed resources for precise reproduction. This tool facilitates access to comprehensive datasets, simplifies the extraction of valuable information, enables metric comparisons, and provides a CLI for easier data access and visualization. As a dynamic resource, ORLB is regularly updated by both its maintainers and the user community, gradually improving the reliability of the available results.

Despite its strengths, ORLB faces challenges in user-friendliness that need to be addressed. Inconsistencies between libraries in evaluation strategies and terminology can make it difficult for users. Scaling community engagement becomes a challenge with more members, libraries, and runs. The lack of Git-like version tracking for runs adds to these limitations.

ORLB is a major step forward in addressing the needs of RL research. It offers a comprehensive, accessible, and collaborative experiment database, enabling precise comparisons and analysis. It improves data access and promotes a deeper understanding of algorithmic performance. While challenges remain, ORLB has the potential to raise the standard of RL research.

---

[6]This section uses the taxonomy introduced by Lynnerup et al. (2019): *repeatability* means accurately duplicating an experiment with source code and random seed availability, *reproducibility* involves redoing an experiment using an existing codebase, and *replicability* aims to achieve similar results independently through algorithm implementation.

## Affiliations

[1]Hugging Face

[2]Drexel University

[3]Univ. Lyon, Centrale Lyon, CNRS, INSA Lyon, UCBL, LIRIS, UMR 5205

[4]SnT, University of Luxembourg

[5]German Aerospace Center (DLR) RMC, Weßling, Germany

[6]Graduate School of System Informatics, Kobe University, Hyogo, Japan

[7]School of Computer Science and Technology, University of Chinese Academy of Sciences

[8]Chengdu Institute of Computer Applications, Chinese Academy of Sciences

[9]University of Maryland, College Park

[10]NVIDIA

[11]Snap Inc.

[12]School of Computer Science, McGill University

[13]Polytech Montpellier DO

[14]Zhejiang University

[15]Department of Computer Science, Purdue University

[16]Work done while at Cohere

[17]Chinese University of Hong Kong, Shenzhen

[18]University College London

[19]Agency for Science, Technology and Research

[20]Faculty of Computer Science, University of Vienna, Vienna, Austria

[21]UniVie Doctoral School Computer Science, University of Vienna

[22]Vidyasirimedhi Institute of Science and Technology (VISTEC)

[23]University of Southampton

[24]Univ. Lille, Inria, CNRS, Centrale Lille, UMR 9189 – CRIStAL

[25]Saint-Gobain Research Paris

[26]International Institute of Information Technology, Hyderabad, India

[27]AIcrowd SA

[28]Valeo Driving Assistance Research

[29]New York University

[30]Sea AI Lab

[31]Institute of Informatics, Federal University of Rio Grande do Sul

[32]AIRI

[33]Department of Automation, Tsinghua University

[34]University of Basel

[35]University of Michigan

[36]UC Berkley

## Acknowledgments

This work has been supported by a highly committed RL community. We have listed all the contributors to date, and would like to thank all future contributors and users in advance.

This work was granted access to the HPC resources of IDRIS under the allocation 2022-[AD011012172R1] made by GENCI. The MORL-Baselines experiments have been conducted on the HPCs of the University of Luxembourg, and of the Vrije Universiteit Brussel. This work was partly supported by the National Key Research and Development Program of China (2023YFB3308601), Science and Technology Service Network Initiative (KFJ-STS-QYZD-2021-21-001), the Talents by Sichuan provincial Party Committee Organization Department, and Chengdu - Chinese Academy of Sciences Science and Technology Cooperation Fund Project (Major Scientific and Technological Innovation Projects). Some experiments are conducted at Stability AI and Hugging Face's cluster.

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

## A  Plotting results guidelines

### A.1  Using the CLI

This section gives notable additional examples of usage of the provided CLI. A more comprehensive set of examples and manual is available in the README page of the project.

#### A.1.1  Plotting episodic return from various libraries

First, we showcase the most basic usage of the CLI, that is comparing two different implementations of the same algorithm based on learning curve of episodic return. For example, Figure 8 and 9 compare CleanRL's TD3 implementation against the original TD3, both in terms of sample efficiency and time. The command used to generate this plot is listed below.

```
python -m openrlbenchmark.rlops \
    --filters '?we=openrlbenchmark&wpn=sfujim-TD3&ceik=env&cen=policy&metric=charts/episodic_return' 'TD3?
        cl=Official TD3' \
    --filters '?we=openrlbenchmark&wpn=cleanrl&ceik=env_id&cen=exp_name&metric=charts/episodic_return' '
        td3_continuous_action_jax?cl=Clean RL TD3' \
    --env-ids HalfCheetah-v2 Walker2d-v2 Hopper-v2 \
    --pc.ncols 3 \
    --pc.ncols-legend 2 \
    --output-filename static/td3_vs_cleanrl \
    --scan-history
```

In the above command, `wpn` denotes the project name, typically the learning library name. This allows to fetch results of implementations from different projects. Moreover, it is possible to specify which metric to compare, in this case `charts/episodic_return`. Also, the CLI provides the possibility to select a given algorithm and apply a different name in the plot, e.g. we rename TD3 to *Official TD3* and `td3_continuous_action_jax` to *Clean RL TD3*. Finally, we can also select a set of environments through the `--end-ids` option.

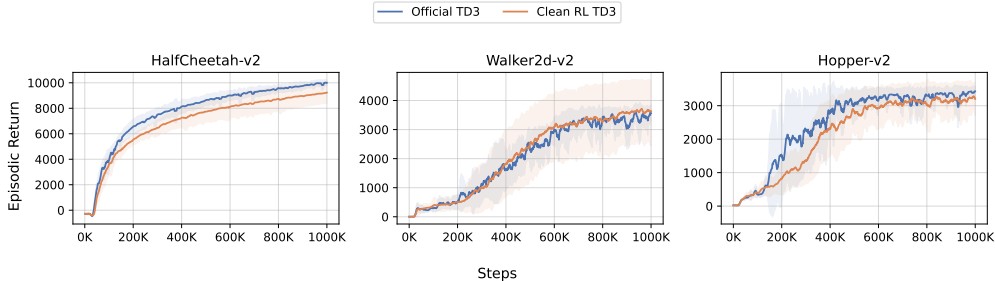

Figure 8: Comparing CleanRL's TD3 against the original TD3 implementation (sample efficiency).

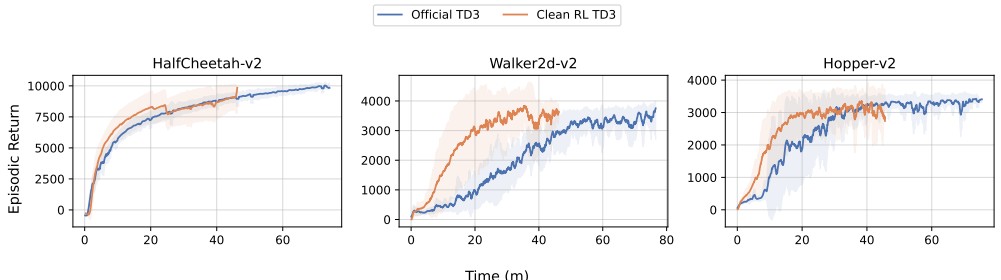

Figure 9: Comparing CleanRL's TD3 against the original TD3 implementation (time).

### A.1.2 RLiable integration

ORLB also integrates with RLiable (Agarwal et al., 2021). To enable such plot, the option `--rliable` can be toggled, then additional parameters are available under `--rc`. Figures 10, 11, 12, 13 showcase the resulting plots of the following command:

```
python -m openrlbenchmark.rlops \
    --filters '?we=openrlbenchmark&wpn=baselines&ceik=env&cen=exp_name&metric=charts/episodic_return' '
        baselines-ppo2-cnn?cl=OpenAI Baselines PPO2' \
    --filters '?we=openrlbenchmark&wpn=envpool-atari&ceik=env_id&cen=exp_name&metric=charts/
        avg_episodic_return' 'ppo_atari_envpool_xla_jax_truncation?cl=CleanRL PPO' \
    --env-ids AlienNoFrameskip-v4 AmidarNoFrameskip-v4 AssaultNoFrameskip-v4 AsterixNoFrameskip-v4
        AsteroidsNoFrameskip-v4 AtlantisNoFrameskip-v4 BankHeistNoFrameskip-v4 BattleZoneNoFrameskip-v4
        BeamRiderNoFrameskip-v4 BerzerkNoFrameskip-v4 BowlingNoFrameskip-v4 BoxingNoFrameskip-v4
        BreakoutNoFrameskip-v4 CentipedeNoFrameskip-v4 ChopperCommandNoFrameskip-v4
        CrazyClimberNoFrameskip-v4 DefenderNoFrameskip-v4 DemonAttackNoFrameskip-v4 DoubleDunkNoFrameskip-
        v4 EnduroNoFrameskip-v4 FishingDerbyNoFrameskip-v4 FreewayNoFrameskip-v4 FrostbiteNoFrameskip-v4
        GopherNoFrameskip-v4 GravitarNoFrameskip-v4 HeroNoFrameskip-v4 IceHockeyNoFrameskip-v4
        PrivateEyeNoFrameskip-v4 QbertNoFrameskip-v4 RiverraidNoFrameskip-v4 RoadRunnerNoFrameskip-v4
        RobotankNoFrameskip-v4 SeaquestNoFrameskip-v4 SkiingNoFrameskip-v4 SolarisNoFrameskip-v4
        SpaceInvadersNoFrameskip-v4 StarGunnerNoFrameskip-v4 SurroundNoFrameskip-v4 TennisNoFrameskip-v4
        TimePilotNoFrameskip-v4 TutankhamNoFrameskip-v4 UpNDownNoFrameskip-v4 VentureNoFrameskip-v4
        VideoPinballNoFrameskip-v4 WizardOfWorNoFrameskip-v4 YarsRevengeNoFrameskip-v4 ZaxxonNoFrameskip-
        v4 JamesbondNoFrameskip-v4 KangarooNoFrameskip-v4 KrullNoFrameskip-v4 KungFuMasterNoFrameskip-v4
        MontezumaRevengeNoFrameskip-v4 MsPacmanNoFrameskip-v4 NameThisGameNoFrameskip-v4
        PhoenixNoFrameskip-v4 PitfallNoFrameskip-v4 PongNoFrameskip-v4 \
    --env-ids Alien-v5 Amidar-v5 Assault-v5 Asterix-v5 Asteroids-v5 Atlantis-v5 BankHeist-v5 BattleZone-v5
        BeamRider-v5 Berzerk-v5 Bowling-v5 Boxing-v5 Breakout-v5 Centipede-v5 ChopperCommand-v5
        CrazyClimber-v5 Defender-v5 DemonAttack-v5 DoubleDunk-v5 Enduro-v5 FishingDerby-v5 Freeway-v5
        Frostbite-v5 Gopher-v5 Gravitar-v5 Hero-v5 IceHockey-v5 PrivateEye-v5 Qbert-v5 Riverraid-v5
        RoadRunner-v5 Robotank-v5 Seaquest-v5 Skiing-v5 Solaris-v5 SpaceInvaders-v5 StarGunner-v5
        Surround-v5 Tennis-v5 TimePilot-v5 Tutankham-v5 UpNDown-v5 Venture-v5 VideoPinball-v5 WizardOfWor-
        v5 YarsRevenge-v5 Zaxxon-v5 Jamesbond-v5 Kangaroo-v5 Krull-v5 KungFuMaster-v5 MontezumaRevenge-v5
        MsPacman-v5 NameThisGame-v5 Phoenix-v5 Pitfall-v5 Pong-v5 \
    --no-check-empty-runs \
    --pc.ncols 5 \
    --pc.ncols-legend 2 \
    --rliable \
    --rc.score_normalization_method atari \
    --rc.normalized_score_threshold 8.0 \
    --rc.sample_efficiency_plots \
    --rc.sample_efficiency_and_walltime_efficiency_method Median \
    --rc.performance_profile_plots \
    --rc.aggregate_metrics_plots \
    --rc.sample_efficiency_num_bootstrap_reps 50000 \
    --rc.performance_profile_num_bootstrap_reps 2000 \
    --rc.interval_estimates_num_bootstrap_reps 2000 \
    --output-filename static/cleanrl_vs_baselines_atari \
    --scan-history
```

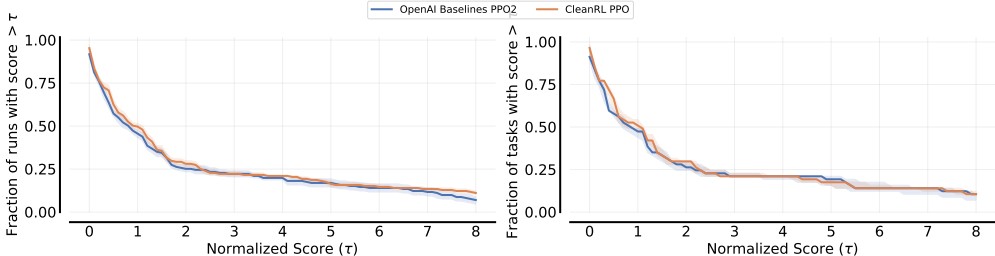

Figure 10: Clean RL PPO vs. OpenAI Baselines PPO, normalized score (RLiable).

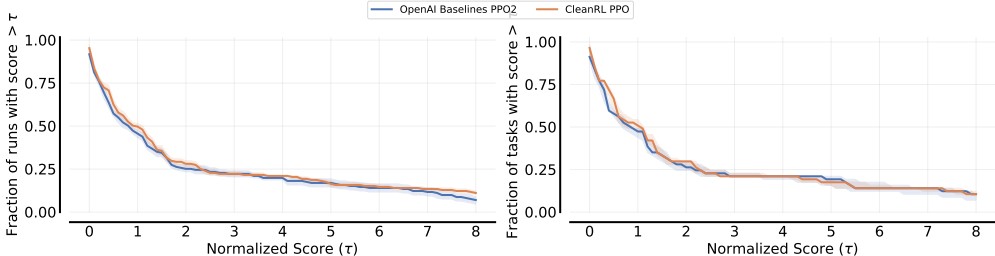

Figure 11: Clean RL PPO vs. OpenAI Baselines PPO, performance profile (RLiable).

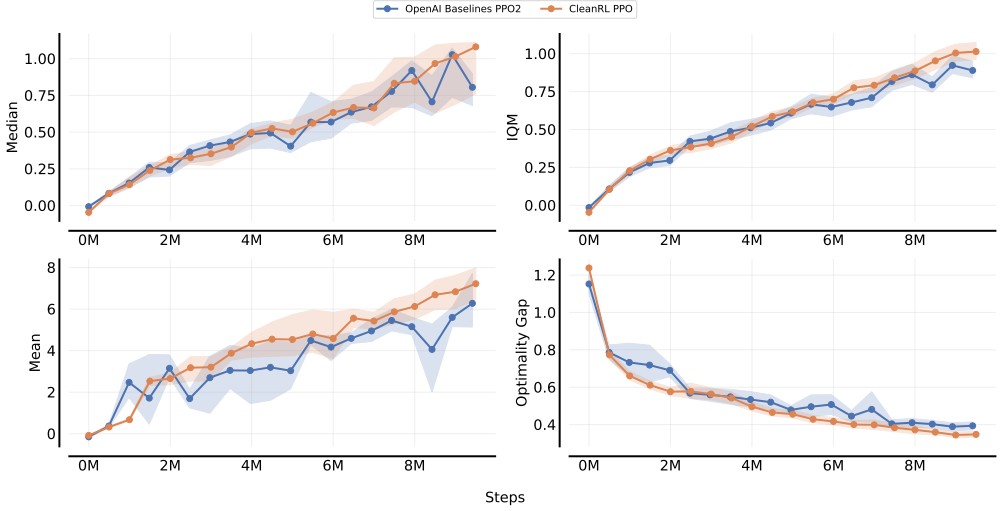

Figure 12: Clean RL PPO vs. OpenAI Baselines PPO, sample efficiency (RLiable).

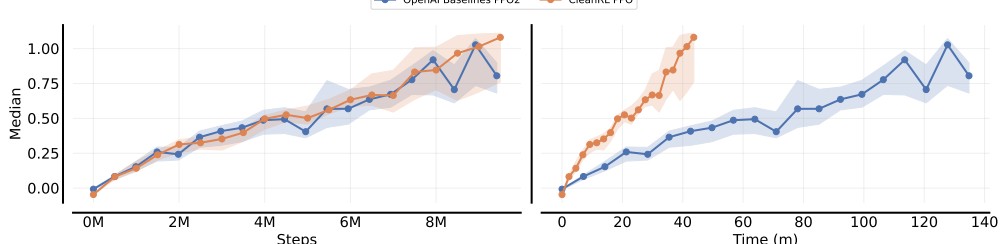

Figure 13: Clean RL PPO vs. OpenAI Baselines PPO, walltime efficiency (RLiable).

### A.1.3 Multi-metrics

Sometimes, such as in multi-objective RL (MORL), it is useful to report multiple metrics in the paper. Hence, the CLI includes an option to plot multiple metrics. Below is an example of CLI and resulting plots (Figure 14) for multiple MORL algorithms on different environments.

```
python -m openrlbenchmark.rlops_multi_metrics \
    --filters '?we=openrlbenchmark&wpn=MORL-Baselines&ceik=env_id&cen=algo&metrics=eval/hypervolume&metrics=
        eval/igd&metrics=eval/sparsity&metrics=eval/mul' \
    'Pareto Q-Learning?cl=Pareto Q-Learning' \
    'MultiPolicy MO Q-Learning?cl=MPMOQL' \
    'MultiPolicy MO Q-Learning (OLS)?cl=MPMOQL (OLS)' \
    'MultiPolicy MO Q-Learning (GPI-LS)?cl=MPMOQL (GPI-LS)' \
    --env-ids deep-sea-treasure-v0 deep-sea-treasure-concave-v0 fruit-tree-v0 \
    --pc.ncols 3 \
    --pc.ncols-legend 4 \
    --pc.xlabel 'Training steps' \
    --pc.ylabel '' \
    --pc.max_steps 400000 \
    --output-filename morl/morl_deterministic_envs \
    --scan-history
```

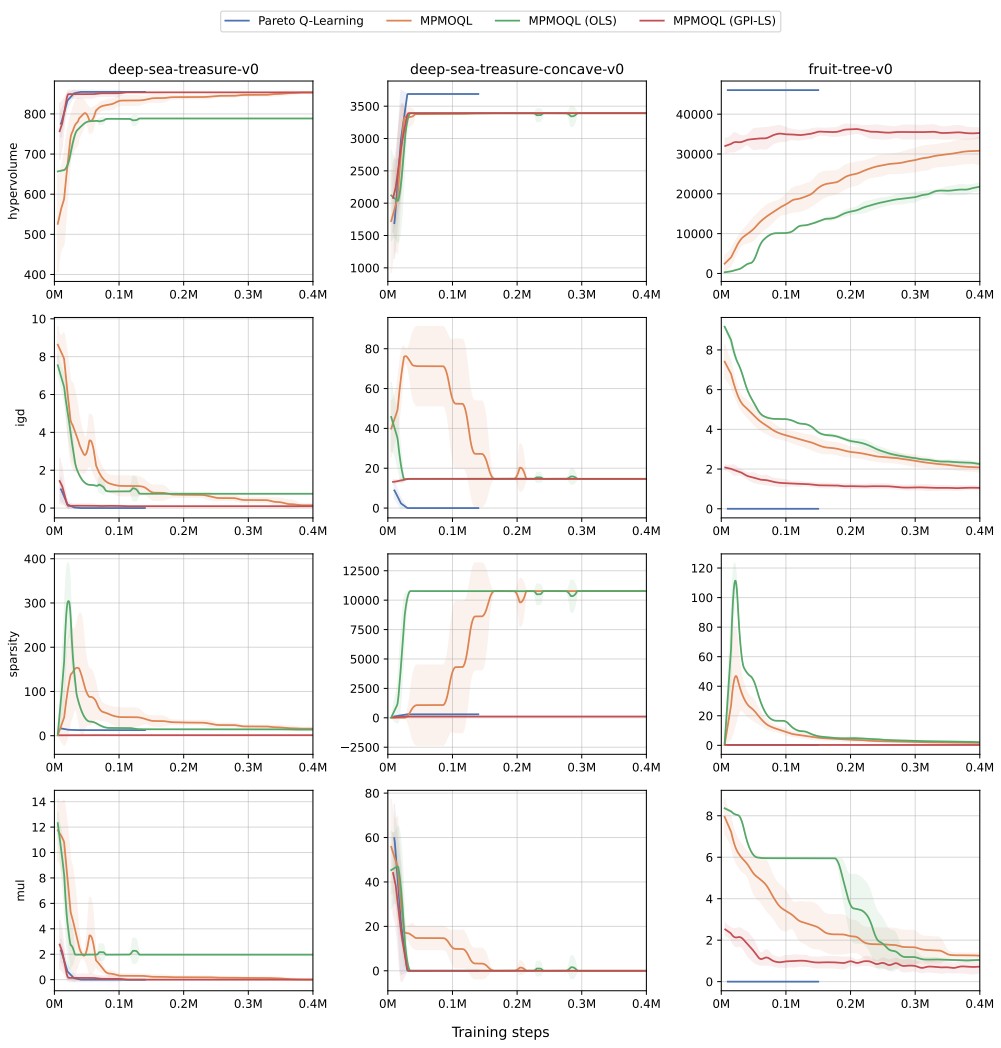

Figure 14: Plotting different metrics for different environments.

## A.2  Using a custom script

Our CLI proves highly beneficial for generating standard RL plots, as demonstrated above. Neverthe-less, in certain specialized cases, researchers may wish to expose the data in an alternative format. Fortunately, all the data hosted in ORLB is accessible through the Weights and Biases API. The following example illustrates how this API can be utilized. From there, researchers can employ any custom script for plotting this data to suit their specific needs. A simple example of such a script is given below, and the corresponding generated plot is shown in Figure 15.

```
import matplotlib.pyplot as plt
import wandb

project_name = "sb3"
run_id = "0a1kqgev"

api = wandb.Api()
run = api.run(f"openrlbenchmark/{project_name}/{run_id}")
history = run.history(keys=["global_step", "eval/mean_reward"])
plt.plot(history["global_step"], history["eval/mean_reward"])
plt.title(run.name)
plt.savefig("custom_plot.png")
```

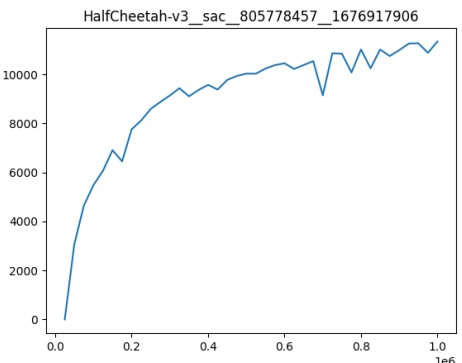

Figure 15: Example of a plot created with a custom script, by importing data directly from ORLB using the WandB API.

# B Additional details for the case study

This appendix gives additional results related to the first case study presented in Section 3.1. Figure 17 shows the results by environment for the Atari benchmark, and Figure 16 shows them for the MuJoCo and Box2d benchmarks. The command lines used to generate these figures are as follows.

```
python -m openrlbenchmark.rlops \
    --filters '?we=openrlbenchmark&wpn=sb3&ceik=env&cen=algo&metric=eval/mean_reward' 'ppo?cl=PPO' \
    --filters '?we=modanesh&wpn=openrlbenchmark&ceik=env&cen=algo&metric=eval/mean_reward' 'ppo?cl=PPO w/
        MC for value estimation' \
    --env-ids BreakoutNoFrameskip-v4 SpaceInvadersNoFrameskip-v4 SeaquestNoFrameskip-v4 EnduroNoFrameskip-
        v4 PongNoFrameskip-v4 QbertNoFrameskip-v4 BeamRiderNoFrameskip-v4 \
    --no-check-empty-runs \
    --pc.ncols 3 \
    --pc.ncols-legend 2 \
    --rliable \
    --rc.score_normalization_method atari \
    --rc.normalized_score_threshold 8.0 \
    --rc.sample_efficiency_plots \
    --rc.sample_efficiency_and_walltime_efficiency_method Median \
    --rc.performance_profile_plots \
    --rc.aggregate_metrics_plots \
    --rc.sample_efficiency_num_bootstrap_reps 1000 \
    --rc.performance_profile_num_bootstrap_reps 1000 \
    --rc.interval_estimates_num_bootstrap_reps 1000 \
    --output-filename static/gae_for_ppo_value_atari_per_env \
    --scan-history \
    --rc.sample_efficiency_figsize 7 4

python -m openrlbenchmark.rlops \
    --filters '?we=openrlbenchmark&wpn=sb3&ceik=env&cen=algo&metric=eval/mean_reward' 'ppo?cl=PPO' \
    --filters '?we=modanesh&wpn=openrlbenchmark&ceik=env&cen=algo&metric=eval/mean_reward' 'ppo?cl=PPO w/
        MC for value estimation' \
    --env-ids InvertedDoublePendulum-v2 InvertedPendulum-v2 Reacher-v2 HalfCheetah-v3 Hopper-v3 Swimmer-v3
        Walker2d-v3 LunarLander-v2 \
    --no-check-empty-runs \
    --pc.ncols 3 \
    --pc.ncols-legend 2 \
    --rliable \
    --rc.normalized_score_threshold 1.0 \
    --rc.sample_efficiency_plots \
    --rc.sample_efficiency_and_walltime_efficiency_method Median \
    --rc.performance_profile_plots \
    --rc.aggregate_metrics_plots \
    --rc.sample_efficiency_num_bootstrap_reps 1000 \
    --rc.performance_profile_num_bootstrap_reps 1000 \
    --rc.interval_estimates_num_bootstrap_reps 1000 \
    --output-filename static/gae_for_ppo_value_mujoco_per_env \
    --scan-history \
    --rc.sample_efficiency_figsize 7 4
```

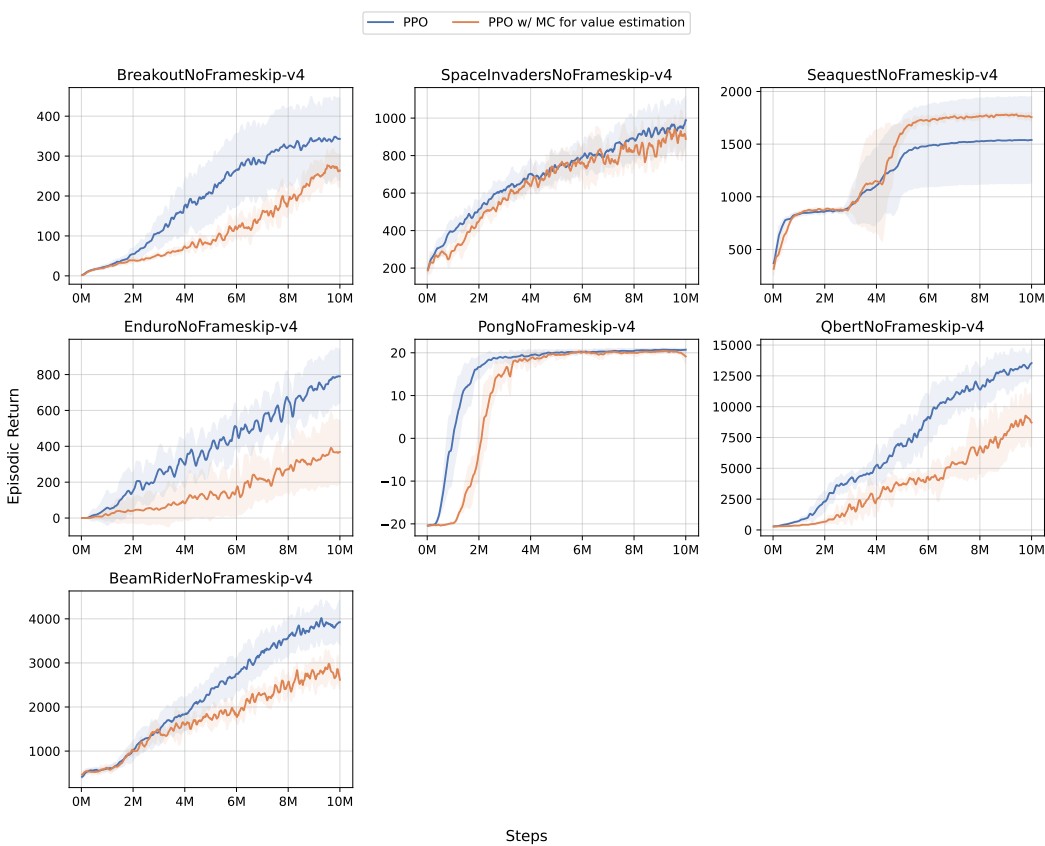

Figure 16: Comparison between the original PPO and the PPO with MC value estimates in various MuJoCo and Box2D environments. Plots represent the evolution of the episodic return as a function of the number of interactions with the environment, and shaded areas represent the standard deviation.

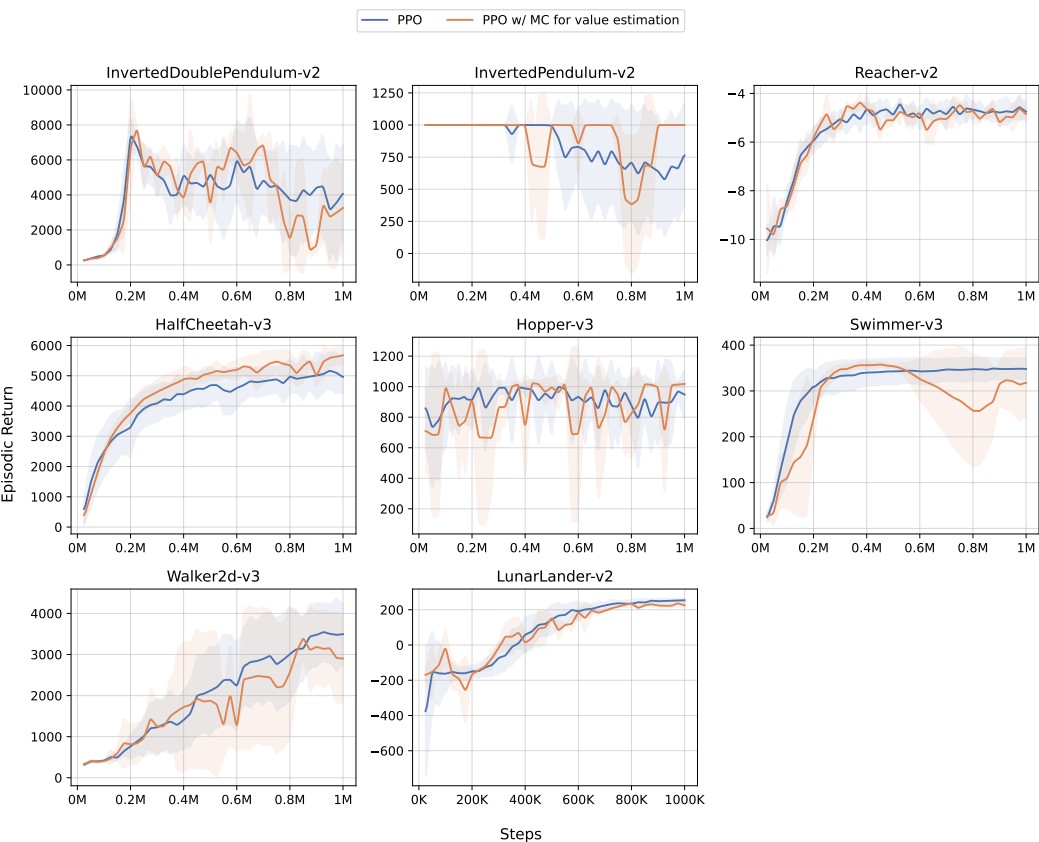

Figure 17: Comparison between the original PPO and the PPO with MC value estimates in various MuJoCo and Box2D environments. Plots represent the evolution of the episodic return as a function of the number of interactions with the environment, and shaded areas represent the standard deviation.

## C   Refine the MuJoCo benchmark with Stable Baselines3

In this appendix, we present a synthetic representation of the learning results of the Stable Baselines3 algorithms (Raffin et al., 2021) tested on the MuJoCo benchmark (Brockman et al., 2016; Todorov et al., 2012), whose data is contained in ORLB. At the time of writing, data from 757 runs has been used, unevenly distributed between the different experiments. It is important to emphasise that the optimisation of hyperparameters and the training budget vary from one experiment to another. Consequently, the results should be interpreted with caution. All the hyperparameters and raw data used to generate these curves are available on ORLB. Figure 18 shows the aggregation of the final performances following the recommendations of Agarwal et al. (2021), and Figure 19 the corresponding performance profiles. Figure 20 shows the learning curves as a function of the number of interactions.

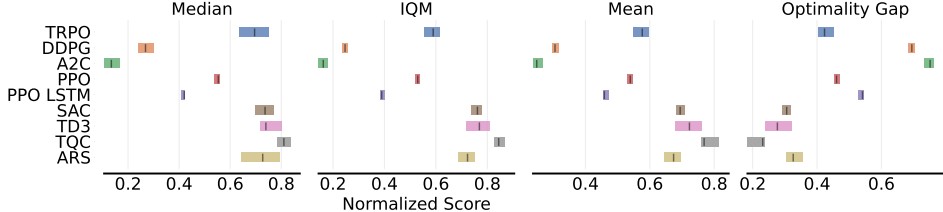

Figure 18: Aggregated final normalized episodic return with 95% stratified bootstrap CIs on the MuJoCo benchmark of the algorithms integrated into Stable Baselines3.

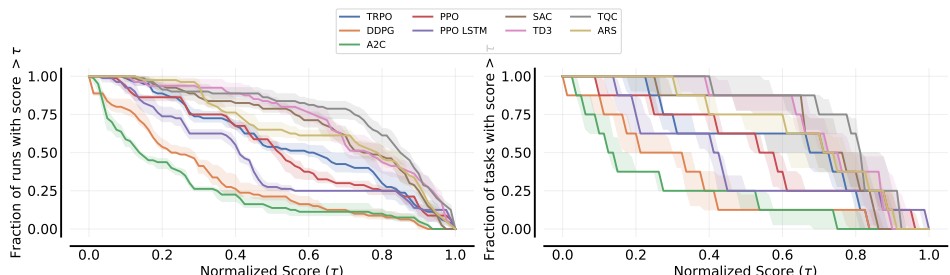

Figure 19: Performance profile of algorithms implemented using Stable Baselines 3 (Raffin et al., 2021) on the MuJoCo benchmark (Todorov et al., 2012). Scores are normalized using the min-max method.

The command used to generate Figures 18, 19 and 20 is as follows[7].

```
python -m openrlbenchmark.rlops \
    --filters '?we=openrlbenchmark&wpn=sb3&ceik=env&cen=algo&metric=eval/mean_reward' 'trpo?cl=TRPO' \
    --filters '?we=openrlbenchmark&wpn=sb3&ceik=env&cen=algo&metric=eval/mean_reward' 'ddpg?cl=DDPG' \
    --filters '?we=openrlbenchmark&wpn=sb3&ceik=env&cen=algo&metric=eval/mean_reward' 'a2c?cl=A2C' \
    --filters '?we=openrlbenchmark&wpn=sb3&ceik=env&cen=algo&metric=eval/mean_reward' 'ppo?cl=PPO' \
    --filters '?we=openrlbenchmark&wpn=sb3&ceik=env&cen=algo&metric=eval/mean_reward' 'ppo_lstm?cl=PPO LSTM
        ' \
    --filters '?we=openrlbenchmark&wpn=sb3&ceik=env&cen=algo&metric=eval/mean_reward' 'sac?cl=SAC' \
    --filters '?we=openrlbenchmark&wpn=sb3&ceik=env&cen=algo&metric=eval/mean_reward' 'td3?cl=TD3' \
    --filters '?we=openrlbenchmark&wpn=sb3&ceik=env&cen=algo&metric=eval/mean_reward' 'ars?cl=ARS' \
    --filters '?we=openrlbenchmark&wpn=sb3&ceik=env&cen=algo&metric=eval/mean_reward' 'tqc?cl=TQC' \
    --env-ids Ant-v3 BipedalWalker-v3 BipedalWalkerHardcore-v3 HalfCheetah-v3 Hopper-v3 Humanoid-v3 Swimmer
        -v3 Walker2d-v3 \
    --no-check-empty-runs \
    --pc.ncols 2 \
    --pc.ncols-legend 4 \
    --rliable \
    --rc.normalized_score_threshold 1.0 \
    --output-filename static/mujoco_sb3 \
    --scan-history
```

---

[7]For Figure 20, we are omitting ARS as it was run with many more steps, and its inclusions hinder readability.

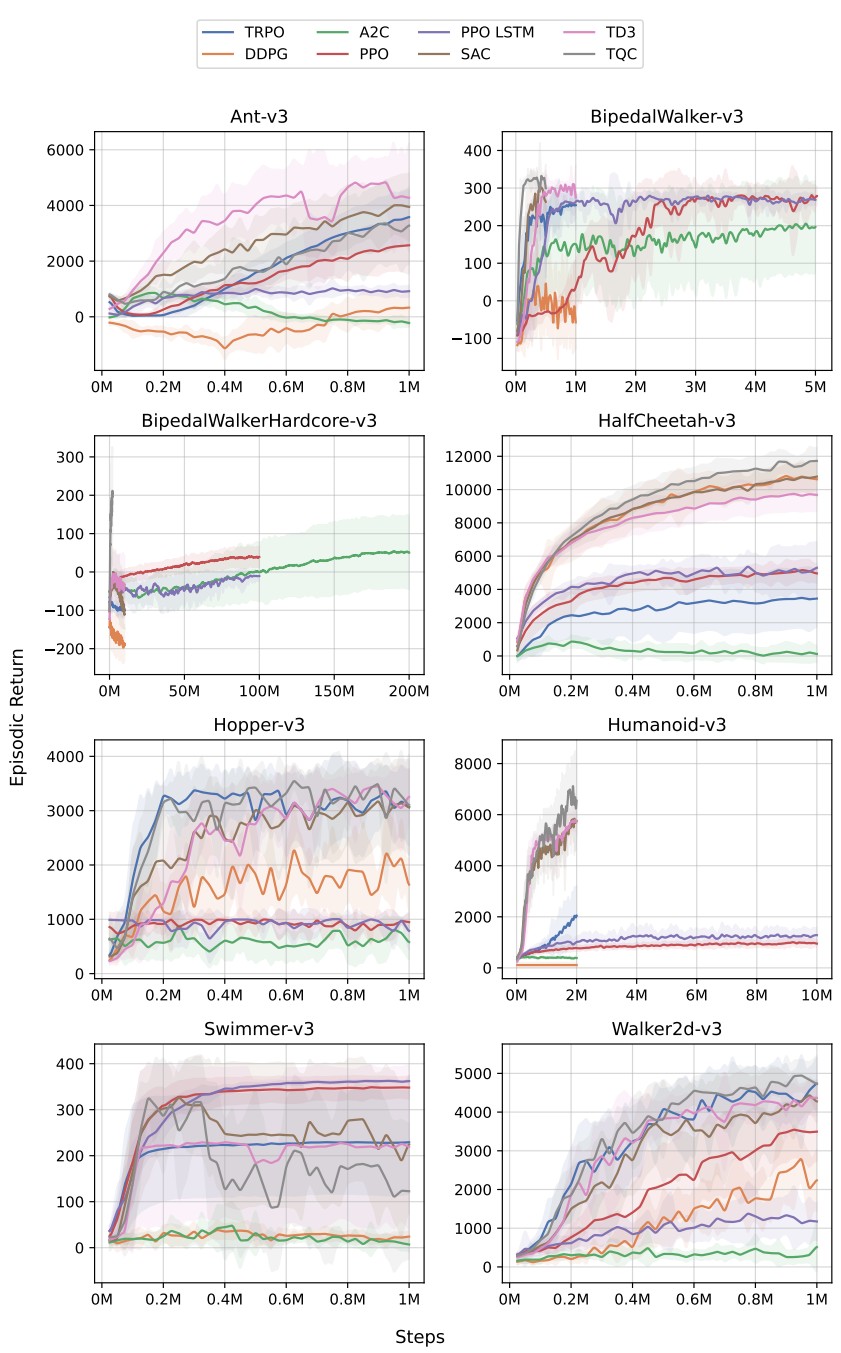

Figure 20: Sample efficiency curves for algorithms on the MuJoCo Benchmark (Todorov et al., 2012). This graph presents the mean episodic return for algorithms implemented using Stable Baselines 3 (Raffin et al., 2021), averaged across a minimum of 10 runs (refer to ORLB for specific run counts). Data points are subsampled to 10,000 and interpolated for clarity. The curves are smoothed using a rolling average with a window size of 100. The shaded regions around each curve indicate the standard deviation.

