# OpenReview forum: "Open RL Benchmark: Comprehensive Tracked Experiments for Reinforcement Learning"
_NeurIPS.cc/2024/Datasets_and_Benchmarks_Track — Submitted to NeurIPS 2024 Track Datasets and Benchmarks_

### Official Review · Reviewer_UaXe · 2024-07-02

**Rating:** 3
**Confidence:** 4

**Review:**

This paper claims six contributions in the introduction section. Among them, "reproducibility" is easy to follow, since the majority of the content tells readers how to obtain or reproduce the results. However, some other contributions are unclear to me. For example, "Extensive dataset": the paper does not provide a new dataset or RL environments but rather a collection of existing testbeds. "Standardization": I did not find new evaluation criteria proposed in the paper. Then, what does this standardization stand for? "Comprehensive metrics": the metrics are briefly discussed in Sec 2.1, which is unclear regarding the specific metrics used. Sec. 4 of the paper ends abruptly. Several inconsistencies of existing methods in experiment settings are discussed. Does ORLB fix them?

**Strengths:**

Tracked running results of well-known RL methods with parameters.

**Additional Feedback:**

N/A

**Clarity:**

The paper uses a plenty of space to describe the usage of ORLB (sec. 3) and pitfalls of other methods (sec. 4), which are not really necessary. Instead, the authors should focus on demonstrating the claimed contributions in Sec. 1.

**Correctness:**

Some claims are questionable (see Review box). The paper does not design new experiments but performs RL methods in existing RL testbeds, thus no major error in evaluation.

**Documentation:**

Though it seems the project link is not mentioned in the paper, I find their github repo: https://github.com/openrlbenchmark/openrlbenchmark . Generally, the instructions for reproducing are described in the repo.

**Opportunities For Improvement:**

Generally, a dataset and benchmark track seeks papers offering new datasets or benchmarking existing methods to provide insights for new algorithm designs. This paper only tracks the running statistics of some well-known RL methods and stores them together. Authors could consider conducting more in-depth comparisons and analyses on the influence of different parameter settings, given that ORLB maintains many running results.

**Relation To Prior Work:**

The related works are discussed in Sec. 5.

**Summary And Contributions:**

The paper introduces open RL benchmark (ORLB). It is a collection of several RL libraries and RL testbeds. In addition, ORLB provides detailed parameter settings and tracked running statistics, e.g., learning curves of various RL algorithms. The motivation of ORLB is to address the reproducibility challenges of RL methods, due to the unclear documentation and software dependencies. The contribution of this paper is thus the tracked running information of different RL libraries in one place.

---

> ### Author Rebuttal · Authors · 2024-08-15
>
> We thank the reviewer for their feedback and suggestions. Below, we provide clarifications to the raised concerns:
>
> > A D&B track seeks papers offering new datasets or benchmarking existing methods to provide insights for new algorithm designs.
>
> We agree with the reviewer that providing insights for new algorithm designs is important, and we strongly believe ORLB provides that. By aggregating tracked metrics from experiments that employ different RL libraries and algorithms, ORLB allows the user to easily compare and evaluate differences in performance coming from different implementations, environments.
>
> A great example of how ORLB has already provided new insights is in the multi-objective RL field. The work of Felten et al., 2023, presented at the last edition of the NeurIPS D&B track, employed ORLB to conduct their benchmark. This dataset hosted on ORLB is widely used by MORL practitioners to compare different MORL algorithms.
>
> We also would like to emphasize that the metrics we have tracked would empower and offer valuable baselines for new practitioners since they can “sanity-check” their results without having to re-implement baseline algorithms from scratch and spend 10,000$+ to reproduce baseline results.
> For example, https://sdpkjc.github.io/snapshotrl/ does exactly that—it re-uses the tracked CleanRL’s SAC metric instead of running experiments from scratch and potentially making mistakes in reproducing the original works. This was previously only possible within big companies such as DeepMind, which have tracked results in their internal servers that are not publicly available.
>
> > Authors could consider conducting more in-depth comparisons and analyses on the influence of different parameter settings
>
> We agree with the reviewer that hyperparameter analyses are valuable. However, this type of analysis (e.g., as done in https://openreview.net/forum?id=nIAxjsniDzg) is outside the scope of our work, and would provide material for an entirely new paper. Our goal with ORLB is to provide strong baselines using already-well-tuned hyperparameters, so future practitioners can compare with strong baselines instead of implementing potentially flawed baselines. This also means many hyperparameters are the same. For example, the PPO from CleanRL, SB3, openai/baselines, and some others all use the same set of hyperparameters for Atari experiments.
> Nevertheless, in case future research performs such a study, ORLB would be the perfect tool to store and democratize the resulting metrics and analyses.
>
> > the paper does not provide a new dataset or RL environments but rather a collection of existing testbeds.
>
> It is true that the aim of ORLB is not to introduce new environments. Rather, it is to provide a large set of training tracking data (‘dataset’), hence we disagree with this being listed as a weakness of our work. Regarding novelty, notice that ORLB is an extensive dataset of at least 72,000 hours worth of tracked runs of popular RL libraries on existing testbeds. It is new because much of the training run data **never existed publicly** on the internet. It's a valuable resource for practitioners, who always struggle to find baseline results, even for popular RL algorithms and libraries, to compare their own results against.
>
> For instance, the OpenAI Baselines library (https://github.com/openai/baselines) has been a foundational RL resource for nearly eight years, but finding specific performance data (let’s say the PPO value loss for MsPacmanNoFrameskip-v4) is challenging. This often forces users to re-implement algorithms themselves, which is error-prone and costly. Unlike other fields, it's been proven that even minor implementation details can significantly influence results in RL. With ORLB, however, you can easily access this critical information, such as the PPO performance data available at https://wandb.ai/openrlbenchmark/baselines/runs/1vldj6yx.
>
> > the metrics are briefly discussed in Sec 2.1, which is unclear regarding the specific metrics used.
>
> Each library tracks different metrics under different names, so it is up to the users to either see the ORLB documentation or check out the documentation of the originating RL library. For example:
>
> * CleanRL documents the details of the tracked metrics here: https://docs.cleanrl.dev/rl-algorithms/ppo/#explanation-of-the-logged-metrics, which matches perfectly with the tracked experiment https://wandb.ai/openrlbenchmark/cleanrl/runs/hvozu5ha
> * SB3 documents them here: https://stable-baselines3.readthedocs.io/en/master/common/logger.html#explanation-of-logger-output which matches perfectly with the tracked experiment https://wandb.ai/openrlbenchmark/sb3/runs/3vv7nyaj
>
> > What does this standardization stand for?
>
> We thank the reviewer for this suggestion and for the room for clarification. A clearer writing is
> - Convenient re-use of existing data: Provides a convenient command line interface (CLI) that helps people obtain existing tracked baseline metrics, reducing the need for re-running baselines and saving computational resources. We will update the paper accordingly.
>
> By standardization, we meant that the tracked runs would always track a variety of things as a standard practice, as explained in Section 2.1. The tracked run always keeps track of the requirements.txt, hyperparameters, the exact command to reproduce results, training and system metrics, and logs.
>
> The specific evaluation criteria are up to the specific library. For example, SB3 always runs a particular evaluation pipeline (for example, using a smaller epsilon with the DQN method), whereas CleanRL performs no such evaluation pipeline and instead only relies on the training results as the implicit evaluation results. Also, see the explanation of tracked SB3 and CleanRL metrics above.
>
> > it seems the project link is not mentioned in the paper
>
> We thank the reviewer for his suggestion and inform them that the link has been added.

---

> > ### Comment · Reviewer_UaXe · 2024-08-15
> > **Incorrect claims and unclear advantages**
> >
> > Thanks to the authors for their response!
> >
> > First, I appreciate the effort put into reproducing the training results of the RL algorithms.
> >
> > However, my main concerns is the clarity of the paper (not good for publish). After reading it, I found it difficult to grasp the contributions mentioned in the introduction and the advantages of this benchmark compared to others.
> >
> > For claims, from the authors' reply, it seems the authors acknowledge that some statements in the introduction may be inaccurate or unclear. For example, there is no "extensive dataset", the "comprehensive metrics" are not clearly described.
> >
> > For advantages, Sec. 4 discussed several inconsistencies of existing methods in experiment settings. Does ORLB fix them? This isn’t explained in the paper, and it’s crucial to highlight how ORLB stands out from other benchmarks. If ORLB does not address these inconsistencies, then why would we have a benchmark still with pitfalls?
> >
> > I encourage the authors to refine their paper to ensure that their claims are supported by evidence and that the advantages of ORLB are clearly highlighted and easy to understand. Additional in-depth analysis would also be beneficial if the authors wish to expand their contributions.

---

> > > ### Author Rebuttal · Authors · 2024-08-16
> > >
> > > We sincerely thank the reviewer for their prompt response.
> > >
> > > > After reading it, I found it difficult to grasp the contributions mentioned in the introduction and the advantages of this benchmark compared to others.
> > >
> > > One of the precise contributions of this paper is that, to our knowledge, there are no experiment tracking datasets that remotely compares to the wealth and kind of data that ORLB contains. Hence, it seems hardly doable to compare to existing experiment tracking datasets. We are happy to include a comparison if the reviewer could point us to other benchmarks.
> > >
> > > > For claims, from the authors' reply, it seems the authors acknowledge that some statements in the introduction may be inaccurate or unclear.
> > >
> > > We acknowledge that some of the wording may lack clarity, and we thank the reviewer for pointing this out to us. We have taken these comments into account and updated the article (see openreview) to improve clarity.
> > >
> > > > For example, there is no "extensive dataset", the "comprehensive metrics" are not clearly described.
> > >
> > > ORLB provides 72,000 hours of tracked experiments, which is the reason we consider it to be an extensive dataset, insofar as it corresponds to a volume that no single contributor was remotely capable of generating on its own.
> > >
> > > Regarding the “comprehensive metrics,” we refer again to our text and reply stating that: “The specific metrics available may vary from one library to another.” Describing the metrics exposed by each library in detail seems cumbersome and not future proofs. This is why we point to the libraries’ documentation instead.
> > >
> > > > For advantages, Sec. 4 discussed several inconsistencies of existing methods in experiment settings. Does ORLB fix them? This isn’t explained in the paper, and it’s crucial to highlight how ORLB stands out from other benchmarks. If ORLB does not address these inconsistencies, then why would we have a benchmark still with pitfalls?
> > >
> > > We would like to clarify again that ORLB is not a benchmark in the traditional sense of providing new RL problem domains for testing algorithms. Instead, ORLB allows RL practitioners to use existing RL algorithms and environments in a transparent way that was not possible before. This is possible since ORLB provides all the necessary information to reproduce existing results, which is a major and recurring challenge in our field. It allows to evaluate new algorithms in a way that is fair and consistent with the state-of-the-art results in our dataset, which consists of 72,000 hours of tracked experiments. Again, no other dataset of this type exists in our community.
> > >
> > > While fixing all challenges regarding experimental design in RL is a long-term and difficult goal in our community, we strongly believe that *ORLB is a big step in this direction*. Below is a detailed answer to each of the points identified in Sec 4.
> > >
> > > **Learning curve practices**
> > > By providing raw data, ORLB solves this problem occurring in research papers. Anyone can manipulate or plot any data that is provided in ORLB for their research.
> > >
> > > **Data sharing**
> > > “ORLB is community-driven: anyone can download, use, and contribute to the data.” The open nature of ORLB makes it easy for anyone to access the raw training data.
> > >
> > > **Reproducibility**
> > > From the paper: “Each experiment includes a complete configuration with all hyperparameters, frozen versions of dependencies, and the exact command, including the necessary random seed, for systematic reproducibility.”
> > > ORLB provides detailed information regarding the hyperparameters, environment versions, algorithms libraries, and all relevant information necessary to make RL experiments reproducible.

---

> > > > ### Comment · Reviewer_UaXe · 2024-08-28
> > > >
> > > > Thanks authors for the reply! I will keep my score and I encourage the authors to refine the paper for clarity.

---

### Official Review · Reviewer_Vf1K · 2024-07-22
**Open RL Benchmark**

**Rating:** 6
**Confidence:** 4
**Correctness:** Yes
**Clarity:** Well written

**Review:**

The quality of the research is high. The authors have meticulously curated a vast dataset with extensive documentation and tools to ensure its usability. The inclusion of detailed experiment parameters and dependency versions significantly enhances the reproducibility of results.
The paper is well-written and clearly outlines the motivation, contributions, and features of ORLB.
The significance of ORLB is substantial. By providing a standardized dataset with comprehensive metrics, ORLB has the potential to accelerate RL research, improve reproducibility, and facilitate better comparisons across different studies. Its community-driven nature also ensures continuous growth and relevance.

Pros:
1. Comprehensive dataset with over 25,000 tracked runs.
2. Practical CLI for easy data access and visualization.
3. Community-driven approach encourages widespread use and contribution.

Cons:
1. The scalability of ORLB seems limited. If a new, previously unrecorded metric needs to be compared, which could be a sum of several tracked metrics, it does not appear to be easily comparable (if this is incorrect, please clarify).
2. The paper could benefit from a more detailed discussion on potential limitations or biases in the dataset.
3. The scalability of the system as the dataset grows could be discussed more thoroughly.

**Strengths:**

The primary strengths of ORLB lie in its comprehensive nature and emphasis on reproducibility. By tracking a wide array of metrics and providing detailed documentation, ORLB ensures that experiments can be precisely replicated, which is crucial for advancing RL research. The community-driven approach not only democratizes access to high-quality data but also fosters collaboration and continuous improvement. The CLI adds a practical tool for researchers, enhancing the usability of the dataset. Overall, ORLB is a significant contribution that aligns well with the broader goals of the RL research community.

**Additional Feedback:**

Question:
1. How do you handle inconsistencies in information recorded by different RL libraries, such as variations in reward and return, or differences between training and testing episode rewards?
2. If I want to compare my algorithm with others using a new metric, which may be entirely new or derived from existing metrics, how should I proceed?

**Documentation:**

Yes

**Opportunities For Improvement:**

See Cons

**Relation To Prior Work:**

Yes

**Summary And Contributions:**

The paper introduces Open RL Benchmark (ORLB), a comprehensive dataset aimed at addressing the lack of standardized metrics and benchmarks in Reinforcement Learning (RL) research. ORLB includes a diverse collection of tracked RL experiments, encompassing not only episodic returns but also algorithm-specific and system metrics.

---

> ### Author Rebuttal · Authors · 2024-08-15
>
> We want to thank the reviewer for their feedback and we appreciate their constructive comments and will address them to strengthen our work.
>
> > How do you handle inconsistencies in information recorded by different RL libraries, such as variations in reward and return, or differences between training and testing episode rewards?
>
> This is definitely a core question, and it touches on ORLB's role in promoting standardization across different RL libraries. ORLB doesn't have the authority to impose a universal standard, but rather aims to remain flexible in accommodating the various metrics used by different libraries. Our approach is to provide clear and detailed documentation that explains how these metrics relate to each other, (e.g. https://github.com/openrlbenchmark/openrlbenchmark?tab=readme-ov-file#currently-supported-libraries), helping users navigate the differences. Moreover, the existence of a cross-library tool like ORLB can encourage library maintainers to move toward a more standardized way of reporting.
>
> >  If a new, previously unrecorded metric needs to be compared, which could be a sum of several tracked metrics, it does not appear to be easily comparable (if this is incorrect, please clarify). If I want to compare my algorithm with others using a new metric, which may be entirely new or derived from existing metrics, how should I proceed?
>
> If it is a completely new and unrecorded metric, then it’s not possible to compare. For example, CleanRL’s DQN does record the metric of average Q-values; however, it’s not possible to obtain the maximum Q-values post hoc because we didn’t record it.
>
> If it is a metric under a different name recording the same thing, ORLB allows you to compare them directly. For example, in the CleanRL vs openai/baselines example, CleanRL’s metric name is `metric=charts/avg_episodic_return` and openai/baselines’s metric name is `metric=charts/episodic_return`. So we can assemble the CLI command as shown in
> https://github.com/openrlbenchmark/openrlbenchmark?tab=readme-ov-file#compare-cleanrls-ppo-with-openaibaseliness-ppo2-on-atari-games. This really applies to arbitrary metrics as well, so people can compare losses under different names, etc.
>
> If this is a metric derived from an existing metric, then it is possible as well. Users can either do this directly via the WandB interface (see the WandB documentation to see how), or fetch existing metrics from the dataset (see Appendix A.2. Using a custom script), so they can sum up several metrics or do whatever suits their research needs.
>
> > The paper could benefit from a more detailed discussion on potential limitations or biases in the dataset.
>
> This is a great suggestion. We have added a limitations section in the paper. For the most part, the limitations include uncontrolled software dependencies and hardware variations across experiments, leading to potential inconsistencies in performance outcomes and runtime comparisons. Additionally, the number of random seeds varies due to different computational budgets, and there is a lack of standardized testing environments across RL libraries. Moreover, there are limited protocols for tracking different versions of RL libraries, potentially affecting the consistency and relevance of the tracked runs. Despite these issues, contributors are encouraged to document dependencies and tag experiments with library versions to mitigate some of these limitations. We invite the reviewer to consult this new section for a detailed study of the limitations.

---

### Official Review · Reviewer_cvk8 · 2024-07-24
**Almost certainly the largest RL benchmark ever conducted. A significant step forward in reporting standards for the RL field.**

**Rating:** 8
**Confidence:** 4
**Correctness:** Yes.
**Clarity:** The paper is well-written and easy to…

**Review:**

The sheer scale of the benchmarking conducted by the authors is remarkable. The level of detail that they recorded for each run sets a new standard for experimental reporting in RL. This work is likely to be very impactful in the broader RL research community because it takes a large and important step toward addressing the reproducibility crisis in the field. By providing the raw logged data for so many runs, the authors have significantly lowered the computational burdon of doing future research in the field, which will help substantially democratize RL research. Furthermore, the report is clear and easy to follow. Overall, this work is a very welcomed contribution to the field and a strong fit for the Datasets and Benchmarking track.

**Strengths:**

The scale of the benchmarking conducted by the authors is the most significant strength of the paper. The WANDB reports created for each set of runs are also a notable contribution as they provide a transparent and easy-to-use summary of each set of runs, and will no doubt be very useful to future researchers. The CLI is another strength that significantly adds to the accessibility of the benchmark results.

**Additional Feedback:**

None.

**Documentation:**

Yes.

**Ethics:**

Not applicable.

**Limitations:**

The authors adequately addressed the limitations of their work in the final section of the report.

**Opportunities For Improvement:**

If space allows in the camera-ready version, consider adding the CLI example in A1.1 to the main text in order to draw greater attention to this valuable contribution.

**Relation To Prior Work:**

Given throughout.

**Summary And Contributions:**

In this work, the authors review the current state of experiment reporting in the field of RL, with a particular focus on learning curves. They highlight that the raw data of learning curves is rarely reported and that this makes it necessary for other researchers to repeatedly attempt to reproduce runs for follow-up research, which is error-prone and computationally expensive. To address this the authors tracked over 25000 RL experiments across a range of environments and algorithm implementations. All experiments are stored on an openly accessible Weights and Biases account and can be accessed and aggregated using a CLI.

---

> ### Author Rebuttal · Authors · 2024-08-15
>
> We thank the reviewer for the positive feedback on our work.
>
> > If space allows in the camera-ready version, consider adding the CLI example in A1.1 to the main text in order to draw greater attention to this valuable contribution.
>
> We would like to thank the reviewer for this very relevant recommendation. If there is space, we will add an example of the CLI in the main text.

---

### Official Review · Reviewer_kfSF · 2024-07-24
**A platform for open sourcing RL experimental results**

**Rating:** 7
**Confidence:** 3

**Review:**

The paper's strengths lie in addressing critical issues of reproducibility and data sharing in RL, offering detailed metrics, tools for exact experiment replication, and a user-friendly CLI for data access and visualisation. The paper is well-motivated by a thorough review of current practices and demonstrates ORLB's utility through case studies. However, the work has notable weaknesses: it lacks a systematic evaluation of ORLB's impact on research productivity or reproducibility rates, provides limited comparative analysis with existing benchmarks, and does not thoroughly address data quality assurance, long-term maintenance challenges, or potential biases in the dataset. Additionally, the paper would benefit from a more comprehensive discussion of limitations, potential negative impacts, and ethical considerations of such a large-scale benchmarking effort.

**Strengths:**

Strengths of the paper are:

- Originality: ORLB appears to be the first large-scale, open benchmark of its kind for RL, addressing a clear need in the field.
- Comprehensiveness: The benchmark covers a wide range of RL libraries, algorithms, and environments.
- Reproducibility: Providing exact configurations and pinned dependencies is a significant step towards addressing reproducibility challenges in RL.
- The CLI tool and case studies demonstrate concrete ways ORLB can accelerate research.
- The open, collaborative nature of ORLB aligns well with principles of open science.
- The paper provides a detailed review of current practices in RL research, motivating the need for ORLB.

**Additional Feedback:**

No more comments.

**Clarity:**

The paper is generally well-written and clear, with effective use of figures and examples. However, the is an opaque background colour fill in Figure 3 that is not present in Figure 2. This should be standardised between figures. Also, the style used for Figure 1 is inconsistent with the style used for later line graphs. A consistent style would improve readability and accessibility.

**Correctness:**

The evaluation methods and experiment designs appear appropriate, but more details on validation procedures would strengthen this aspect.

**Documentation:**

The paper provides good documentation of ORLB's features and usage, but could benefit from more details on data formats and integration with other tools.

**Ethics:**

The paper does not explicitly discuss ethical considerations. Given the potential impact on RL research practices, a statement on ethical implications would be appropriate, particularly around sharing data on potentially sensitive applications or data.

**Limitations:**

The paper acknowledges some limitations of ORLB, such as inconsistencies between libraries and challenges in scaling community engagement. However, it does not thoroughly address potential negative impacts. For instance, there is no discussion of how ORLB might inadvertently bias research towards certain types of problems or algorithms, or how it might impact the diversity of approaches in RL research. The paper would benefit from a more thorough consideration of these potential limitations and impacts.

**Opportunities For Improvement:**

- While case studies are provided, there is no systematic evaluation of ORLB's impact on research productivity or reproducibility rates.
- The paper does not thoroughly compare ORLB to existing benchmark datasets or tools in RL or other fields.
- There is limited discussion of how data quality and correctness are ensured across the diverse set of experiments.
- Maintenance challenges: The long-term sustainability and update mechanisms for ORLB are not thoroughly addressed.
- The paper does not discuss potential biases in the types of algorithms or environments represented in ORLB. This is not something the authors can solve with guarantees, but some thoughts on how they can try to mitigate this would be useful.
- There is little mention of how ORLB handles or encourages reporting of negative results. Will there be a reward system or something to incentivise this?

**Relation To Prior Work:**

A more systematic comparison to prior benchmarking efforts would be valuable, although this is not grounds for rejection.

**Summary And Contributions:**

The authors introduce a comprehensive, open-source dataset and toolset for reinforcement learning research, addressing challenges in reproducibility, data sharing, and benchmarking. The aim is to accelerate progress in the field by enabling more precise comparisons between methods and facilitating access to benchmark data across multiple algorithms, libraries and environments.

Contributions:
- A large collection of tracked RL experiments across multiple algorithms, libraries, and environments, totaling over 25,000 runs.
- Detailed tracking of metrics beyond just episodic return, including algorithm-specific and system metrics.
- Tools for reproducing experiments exactly, including full configurations and dependency versions.

---

> ### Author Rebuttal · Authors · 2024-08-15
>
> We thank the reviewer for taking the time to review our work. We are very excited to share this dataset and toolchain with the reinforcement learning community, so more people can benefit from open access to RL training metrics from different libraries. Regarding opportunities for improvements:
>
> > The paper does not thoroughly compare ORLB to existing benchmark datasets or tools in RL or other fields.
>
> We would love to make such comparisons, but it’s difficult to find existing benchmark datasets or tools. In fact, this was the main motivation for the project. Most benchmark papers only provide a table of the final results, with no openly accessible learning curves, tracked runs, or reproducibility information (e.g., https://proceedings.mlr.press/v48/duan16.pdf)
> As far as other fields are concerned, this is an intriguing question. To our knowledge, no similar initiative exists in supervised or unsupervised/semi-supervised learning due to their relatively stable nature. However, looking at the broader context of open-source practices, there has been significant progress in sharing trained models/checkpoints and datasets. We welcome and encourage this trend toward more open accessible metrics.
>
> > There is limited discussion of how data quality and correctness are ensured across the diverse set of experiments.
>
> This is a good point. We have added a limitation section that explains the data quality in more detail. We noted that the software dependencies and hardware are not controlled. For example, different RL libraries may have used slightly different Atari software versions, which may implement BreakoutNoFrameskip-v4 slightly differently. However, for the most part, all BreakoutNoFrameskip-v4 experiments are comparable.
>
>
>
> > Maintenance challenges: The long-term sustainability and update mechanisms for ORLB are not thoroughly addressed.
>
> We agree this is a potential limitation of our work. Regarding the update mechanisms, we encourage the contributing members to tag the tracked experiments, so they can use ORLB to keep track of different versions of their libraries, such as done here https://github.com/vwxyzjn/cleanrl/pull/424#issuecomment-1801949569
>
> > The paper does not discuss potential biases in the types of algorithms or environments represented in ORLB. This is not something the authors can solve with guarantees, but some thoughts on how they can try to mitigate this would be useful.
>
> This is a good point. As we have explained in our new limitation section, potential biases include software dependencies, which could cause very subtle performance differences. E.g., https://github.com/openrlbenchmark/openrlbenchmark?tab=readme-ov-file#more-examples  shows that CleanRL’s PPO has the very similar looking learning curves as openai/baselines’ PPO in 53/57 tasks, but there are 4 learning curves look suspiciously different (e.g., YarsRevengeNoFrameSkip-v4). It’s hard to identify the exact reasons for the difference, but the tracked experiments always record the set of dependencies (e.g., https://wandb.ai/openrlbenchmark/cleanrl/runs/178zblge/files/requirements.txt), so the users can always look up.
>
> > There is little mention of how ORLB handles or encourages reporting of negative results. Will there be a reward system or something to incentivise this?
>
> We don’t specifically encourage people to report negative results, but we generally encourage people to track as much as possible, and we do have many negative results. For example, PPO fails on MountainCar-v0 completely (https://docs.cleanrl.dev/rl-algorithms/ppo/#experiment-results). In comparison, some DQN runs could solve MountainCar-v0 (https://docs.cleanrl.dev/rl-algorithms/dqn/#experiment-results_1) and some would fail (this is why we require people to at least track 3 random seeds, so there we can see some variability of the results)

---

### Author Response · Authors · 2024-08-15
**Limitation section**

Dear reviewers,

We have added a limitation section to the paper. As NeurIPS does not allow us to upload a revision of the paper, we put the section below.



While ORLB contains tracked experiments from a variety of RL libraries and environments, limitations apply, and the users should take notes:

1. **Uncontrolled software dependencies**: The tracked experiments were likely conducted using different software dependencies. This means we cannot guarantee perfect controlled experiments. Using the Atari software dependency as an example, OpenAI Baselines uses the deprecated `atari-py==0.2.6`, whereas CleanRL uses `envpool==0.6.4`. While both dependencies implement `BreakoutNoFrameskip-v4` largely the same to the best of our knowledge, there may be subtle changes in this Atari software that cause bias in the performance. In particular, CleanRL's PPO's learning curves are notably different in 4 out of 57 tasks[^1]
   - While this is a possible issue, environment software maintainers mostly keep good backward compatibility between environment versions, so people can expect `BreakoutNoFrameskip-v4` to be the same, but `ALE/Breakout-v5` to be more different.
   - Furthermore, each tracked experiment would also record the set of dependencies automatically (e.g., https://wandb.ai/openrlbenchmark/cleanrl/runs/c1y1qnz4/files/requirements.txt), so users can always double check.

2. **Uncontrolled hardware**: Ultimately, ORLB is a community-based project, meaning different contributing members have used a variety of machines with different GPUs and CPUs. This means that users should only use the runtime information as a reference because they may get different runtimes on different hardware.

3. **Different number of random seeds**: ORLB does not enforce the number of random seeds conducted for the experiments because community members have different computational budgets and preferences. That said, ORLB generally requires the experiments to have at least three random seeds of the same hyperparameters, so at least the users have some sense of the variabilities between runs.

4. **Limited standardized testing environments**: ORLB has limited standardized testing environments across all RL libraries. Ideally, we would like all RL libraries to contribute tracked experiments for 57 Atari games, 9 MuJoCo tasks, and others. In practice, this is difficult for several reasons: 1) not all libraries support all standardized testing environments (e.g., the TD3 repo `sfujim/TD3` does not support Atari games; 2) the community members have different computational budgets. This means the users would need to rely on our docs to find relevant experiments to do analysis.

5. **Limited protocols for controlling RL library versions**: ORLB has limited protocols for tracking runs of the same library of different versions; it also offers no guarantee that the tracked runs are from the latest RL library version, largely due to computational constraints.
   - That said, we do encourage the contributing members to tag experiments with RL library versions[^2], so they can perform regression analysis. Furthermore, we encourage the members to track experiments, should there be significant breaking changes.

* [^1]: See https://github.com/openrlbenchmark/openrlbenchmark?tab=readme-ov-file#more-examples
* [^2]: See https://github.com/vwxyzjn/cleanrl/pull/424#issuecomment-1801949569

---

### Decision · Program_Chairs · 2024-09-26

**Decision:**

Reject

**Comment:**

**Summary of Contributions**

- This paper introduces a comprehensive, open-source dataset for reinforcement learning, including a large collection of RL experiments (over 25,000 tracked runs) across algorithms, libraries, and environments. This paper also provided detailed tracking of metrics and tools for reproducing experiments exactly.

**Strengths**

- The benchmark covers a wide range of RL libraries, algorithms, and environments.
- Community-driven approach encourages widespread use and contribution.

**Weaknesses**

- This work lacks addressing data quality assurance, long-term maintenance challenges, or potential biases in the dataset.
- There is no discussion on how it might impact the diversity of approaches in RL research.
- Reviewer UaXe questioned the clarity of the paper and claimed that some statements in the introduction are inaccurate or unclear, thus, giving the clear rejection rating. The authors also admitted the unclear writing in their responses. For example, the contribution of "extensive dataset", and the "comprehensive metrics" are not clearly described.

**Discussion of Reviewers' Opinions**

- There is hardly any feedback though I have tried my best to encourage the discussion. Reviewer UaXe emphasized his concerns on the clarity and other reviewers had no comments.

**Final Recommendation**

- After carefully considering the reviews and my own evaluation, regarding the clarity, which is the main concern, I recommend rejecting this paper. The author needs to refine the paper for clarity.